# Pixel Self-Attention Guided Real-Time Instance Segmentation for Group Raised Pigs

**DOI:** 10.3390/ani13233591

**Published:** 2023-11-21

**Authors:** Zongwei Jia, Zhichuan Wang, Chenyu Zhao, Ningning Zhang, Xinyue Wen, Zhiwei Hu

**Affiliations:** College of Information Science and Engineering, Shanxi Agricultural University, Jinzhong 030801, China; wzc0847@163.com (Z.W.); 17835714741@163.com (C.Z.); 18234610986@163.com (N.Z.); 18536494976@163.com (X.W.); zhiweihu@whu.edu.cn (Z.H.)

**Keywords:** instance segmentation, attention mechanism, channel self-attention, spatial self-attention

## Abstract

**Simple Summary:**

Research on real-time instance segmentation technology in the field of pigs is of great significance, as it can provide accurate pig behavior monitoring and management for the animal husbandry industry, and improve breeding efficiency and animal welfare. In this study, we developed a novel pixel self-attention (PSA) module by combining channel attention and spatial attention, which can integrate into the feature pyramid module of the YOLACT model. In addition, we visually analyzed prototype vectors within the YOLACT model, the attention maps obtained using the PSA attention module, and the model’s robustness across different shooting angles and times. The experimental results indicated that the proposed attention module can improve model performance and transferability. Our model provides a new research perspective for the pig instance segmentation task, can initially provide information on individual parts of pigs, and can provide a useful reference for subsequent pig tracking management.

**Abstract:**

Instance segmentation is crucial to modern agriculture and the management of pig farms. In practical farming environments, challenges arise due to the mutual adhesion, occlusion, and dynamic changes in body posture among pigs, making accurate segmentation of multiple target pigs complex. To address these challenges, we conducted experiments using video data captured from varying angles and non-fixed lenses. We selected 45 pigs aged between 20 and 105 days from eight pens as research subjects. Among these, 1917 images were meticulously labeled, with 959 images designated for the training set, 192 for validation, and 766 for testing. To enhance feature utilization and address limitations in the fusion process between bottom-up and top-down feature maps within the feature pyramid network (FPN) module of the YOLACT model, we propose a pixel self-attention (PSA) module, incorporating joint channel and spatial attention. The PSA module seamlessly integrates into multiple stages of the FPN feature extraction within the YOLACT model. We utilized ResNet50 and ResNet101 as backbone networks and compared performance metrics, including AP_0.5_, AP_0.75_, AP_0.5-0.95_, and AR_0.5-0.95_, between the YOLACT model with the PSA module and YOLACT models equipped with BAM, CBAM, and SCSE attention modules. Experimental results indicated that the PSA attention module outperforms BAM, CBAM, and SCSE, regardless of the selected backbone network. In particular, when employing ResNet101 as the backbone network, integrating the PSA module yields a 2.7% improvement over no attention, 2.3% over BAM, 2.4% over CBAM, and 2.1% over SCSE across the AP_0.5-0.95_ metric. We visualized prototype masks within YOLACT to elucidate the model’s mechanism. Furthermore, we visualized the PSA attention to confirm its ability to capture valuable pig-related information. Additionally, we validated the transfer performance of our model on a top-down view dataset, affirming the robustness of the YOLACT model with the PSA module.

## 1. Introduction

The robust development of pig farming holds paramount significance in the context of national food security. Group breeding is one of the main methods of standardized and large-scale pig breeding. However, in group-raising scenarios, the convergence and proximity of live pigs, along with obstructions and alterations in their posture, present formidable challenges for manual recording and monitoring. Computer vision-based automatic analysis technology offers a cost-effective, non-invasive monitoring solution, providing the technical foundation for precise individual pig care. A pivotal challenge within this framework revolves around isolating adherent pig individuals from the complex background.

Research on individual pig segmentation using machine vision has significantly advanced areas across various domains. Conventional methods for pig image segmentation can be broadly categorized into static image segmentation and dynamic image segmentation. Static segmentation methods encompass well-established approaches, such as threshold segmentation [1], edge detection segmentation, watershed transformation [2], and morphological segmentation. On the other hand, dynamic image segmentation methods involve using techniques such as the optical flow method, frame difference method, and background difference method. However, these methods confront two main challenges: (1) They necessitate the manual selection of a large number of feature points for subsequent feature extraction, but human involvement introduces errors or omissions. (2) They only consider simple scenarios wherein large differences are observed between the pig foreground and background. These methods are less effective in complex scenarios that are more representative of practical production environments, such as pigs adhering to one another and obscured by debris.

Deep learning has ushered in transformative opportunities for pig target segmentation, particularly by adopting an end-to-end design approach represented by convolutional neural networks (CNNs), attention mechanisms, and analogous techniques. These approaches obviate the laborious feature engineering and selection intrinsic to traditional machine vision research, which is a superiority substantiated by a wealth of research outcomes. In the investigation of individual pigs, convolutional neural networks are applied in diverse tasks, including pig counting [3,4], pig face recognition [5,6], multi-target tracking [7], pig detection [8,9], recognition of pig behaviors [10], and related endeavors [11,12]. For pig segmentation, Yang et al. [13] first used a fully convolutional network (FCN) to segment lactating sows from top-view images, subsequently refining the FCN’s coarse output through Otsu’s thresholding based on color information such as hue and saturation. Building upon FCNs, Hu et al. [14] proposed a series of algorithms to address pig image segmentation challenges. These algorithms integrated channel and spatial attention blocks into the decoder of UNet and LinkNet [15] network structures, amplifying long-distance dependency relationships and enhancing individual pig part information while attenuating background information such as pigsties. These enhancements significantly improved performance. However, the CNN-based segmentation methods discussed earlier are effective for segregating individual pigs from the farming environment but cannot be used to differentiate between distinct individuals within a single image. Instance-level segmentation, which effectively distinguishes individuals within the same object, is particularly relevant for downstream tasks such as pig target tracking and identity-based motion statistics, endowing it with broader applications. The mask R-CNN model introduced by He et al. [16] is a remarkable model capable of achieving high-precision instance segmentation on adherent pig images, registering substantial success in tasks concerning pig instance segmentation. Gan et al. [17] adopted an anchor-free approach for the instance segmentation of individual sows and piglets. Their methodology involved first localizing the piglet head region to extract features from a region of interest (ROI), followed by attention graph convolution to predict piglet contours and masks. This innovative approach facilitated the instance-level detection of piglet suckling behavior. Tu et al. [18] employed the mask-scoring R-CNN [19] framework to segregate and recognize adherent pig regions within group pig images. This approach entailed three steps: leveraging a residual network and feature pyramid network for feature extraction, generating regions of interest using a region candidate network, and constructing regression, classification, and segmentation branches for instance segmentation. Huang et al. [20] pioneered a single-pixel-based method that employed an enhanced central clustering network to predict object centers. Subsequent mask segmentation was utilized to track these centers, culminating in instance segmentation of individual mothers and piglets. The algorithm’s efficacy was rigorously validated through diverse ablation experiments. Wang et al. [21] crafted a pig detection and segmentation model that amalgamated HRNet [22] and feature pyramid network (FPN) [16] modules with the cascade-mask R-CNN [23] as the baseline network. They introduced a collaborative attention mechanism to tackle mutual occlusion and pig body adherence issues, resulting in high-precision detection and segmentation of group pig images within complex environments. Gao et al. [24] proposed an effective individual instance segmentation of group-housed piglets in a pigsty setting anchored in the mask R-CNN framework. This approach incorporated a dual-pyramid feature extraction network above the fundamental feature extraction network, ultimately delivering location and classification outcomes for detection targets through three branches: category, regression, and mask. Consequently, they achieved effective individual instance segmentation of regions where group-housed piglets in pigsty scenarios are in close proximity to each other. However, the aforementioned mask R-CNN-based methodologies only apply the original mask R-CNN model to individual pigs without introducing substantive alterations to the model structure. In the breeding milieu, nuanced variations in the body characteristics of group pigs are evident, necessitating a more efficient algorithm to differentiate individual pigs.

The attention mechanism is crucial for enhancing the relevance of regional information conducive to instance segmentation while suppressing minor details to improve the model’s effectiveness in feature filtration. It has demonstrated success across various domains and can be harnessed to strengthen the distinctive features of pigs, particularly in intricate areas such as pig noses and hooves. Integrating the attention mechanism in these areas can substantially refine segmentation boundaries. Prominent attention modules such as the convolutional block attention module (CBAM) [25], bottleneck attention module (BAM) [26], and spatial and channel squeeze and excitation (SCSE) [27] operate on feature maps in both channel and spatial dimensions. These modules are applied in the domain of instance segmentation for live pigs. Hu et al. [28] introduced two distinct types of attention blocks into the FPN, encoding semantic interdependencies in channels and spatial attention separately. Task networks such as mask R-CNN [16], cascade mask R-CNN [23], mask scoring R-CNN [19], and HTC [29], paired with ResNet50 and ResNet101 as backbone networks, have been employed in various combinations to achieve instance segmentation of individual pigs in group breeding scenarios. Previous studies have not applied attention mechanisms to pig instance segmentation tasks. However, the approaches mentioned above have three notable limitations: (1) When incorporating both channel and spatial attention information, a single attention filtering process is applied to the final output feature map of the feature extraction network. This neglects the extraction of attention information tailored to different receptive field sizes within the feature extraction network. Given the variation in receptive field sizes and the regions of interest, distinct attention filtering can significantly enhance the segmentation accuracy of targets of varying sizes. (2) Existing instance segmentation methods for individual pigs prioritize enhancing segmentation accuracy, often resulting in large model sizes that are challenging to deploy in practical production environments. Real-time segmentation is often necessary in practice, and the aforementioned methods fall short of meeting this demand. (3) Monitoring individual pigs demands all-weather attributes. The illumination intensity, image quality, and noise ratio can significantly differ between daytime and nighttime scene images. Ensuring the model’s robustness in extracting key points under day–night transformation conditions presents a complex challenge. However, existing methods focus on group-raised pigs within a single data source scenario and do not facilitate instance segmentation of pigs in multi-angle and all-weather scenes. The study of pig instance segmentation is of great significance and can provide effective management tools for the animal husbandry industry. By accurately identifying and segmenting individuals in pig images, researchers and farmers can monitor pig behavior, health, and growth in real time. This helps to improve the production efficiency of animal husbandry, reduce the risk of disease transmission, and promote the sustainable development of animal husbandry. The application of pig instance segmentation technology can also be extended to the field of smart agriculture, introducing advanced digital management methods to the breeding industry, thereby promoting the modernization and intelligent development of animal husbandry.

We present an innovative approach by integrating the pixel self-attention (PSA) module into the YOLACT network architecture and evaluating its model transfer performance across diverse perspectives and lighting conditions. The primary contributions can be summarized as follows:We developed the PSA module by combining channel attention and spatial attention information and seamlessly integrated the module into the YOLACT model by employing ResNet50 and ResNet101 as backbone networks. A comprehensive comparative analysis of its performance in instance segmentation and object detection was conducted against the BAM, CBAM, and SCSE attention modules.We visualized the prototype vectors within the YOLACT model, performing a model-level validation of the final prediction results and the interpretability of the model.Attention feature maps of the PSA attention module were visually represented, providing an intuitive depiction of the attention information acquired by PSA.We transferred the YOLACT model integrated with PSA modules, initially trained with a horizontal perspective, to assess the model’s transferability and robustness across top-down perspectives and nighttime scene data.

## 2. Materials and Methods

### 2.1. Data Collection

We employed two data collection methods: a horizontal perspective and a top-down view. The horizontal perspective data served as the training, validation, and testing sets and were obtained through fieldwork. In contrast, data obtained through the top-down view method exclusively constituted the testing set, assessing the model’s transferability trained on the horizontal perspective dataset. This particular dataset was directly sourced from [30]. It includes data samples captured during the daytime and nighttime, offering diverse scenarios to evaluate the model’s robustness. Figure 1 illustrates the horizontal perspective and top-down view collection methods as (a) and (b), respectively.

Considering that the top-down dataset originates directly from a third-party paper, we emphasize elaborating our data collection method utilizing a horizontal moving perspective. Traditionally, cameras are typically mounted above pigpens to capture data. However, this top-down collection method has two primary limitations: (1) The captured images primarily encompass the pig’s dorsal region, lacking crucial detail in areas such as the face and hooves. These regions contain valuable biological information essential for assessing the health and well-being of the pigs. (2) Furthermore, the fixed lens collection method imposes a limited distance range between the pig and the lens. Models trained on data from fixed lenses are not adaptable to mobile applications and exhibit limited scalability. To mitigate these two drawbacks, we introduced an innovative data collection approach that captures images from a horizontal perspective while allowing for lens position adjustments. Figure 1a illustrates the data collection platform at Shanxi Agricultural University’s Experimental Animal Management Center. We selected 45 pigs aged between 20 and 105 days from 8 pens as research subjects. Addressing the challenges posed by group-raised pigs, characterized by unpredictable movement trajectories, dispersion, and a tendency to cluster, we varied the camera-to-pigpen distance from 0 m to 0.3 m to accommodate different pig sizes. The camera tripod was fixed at heights between 0.5 m and 1.3 m above the ground to obtain images from various horizontal angles.

### 2.2. Data Preprocessing

We used the collected videos to create the experimental dataset for instance segmentation in group pigs through two procedures:
We segmented the collected video data by extracting frames every 25 frames. The resulting image resolution was 1920 × 1080, subsequently adjusted to 2048 × 1024 to maintain a 2:1 aspect ratio. Any excess edge regions were filled with white pixels. Data labeling was conducted using LabelMe (http://labelme.csail.mit.edu/Release3.0/) (accessed on 1 August 2023). To reduce the memory requirements of the model, we scaled down the annotation data to 512 × 256, resulting in an initial annotation dataset comprising 1917 images. This dataset was divided into 959 images for training, 192 for validation, and 766 for testing.We applied data augmentation during training to the initial annotated dataset to introduce greater diversity into the image dataset. Instead of generating augmented images using a fixed approach before training, we dynamically augmented the data during the training process based on predefined probabilities. The advantages of in-training data augmentation include generating more random data and subjecting each image to multiple augmentation operations with specified probabilities in each training iteration. These are improvements over fixed pre-augmentation strategies that limit the diversity of image data. Table 1 outlines the augmentation methods employed during in-training data augmentation, their corresponding parameters, and the probabilities of augmentation operations.

## 3. Real-Time Instance Segmentation Model

### 3.1. YOLACT Basic Model

You Only Look At Coefficients (YOLACT) [31] is a deep learning model meticulously designed for real-time object detection and instance segmentation. Its inception in 2019 ushered in a one-stage detector proficient in seamlessly combining object detection and instance segmentation tasks, all while upholding real-time inference capabilities and high precision. Refer to Figure 2 for an illustrative representation of the model’s structure. YOLACT comprises the following essential sub-modules:**Backbone Network:** YOLACT employs well-established networks like ResNet50 [32] or ResNet101 [32] as its backbone. This network component extracts image features, which will be harnessed for object detection and segmentation in subsequent modules. The experimental section presents a comprehensive evaluation of different backbone networks.**FPN (Feature Pyramid Network):** The FPN is used to extract semantic information from feature maps of diverse scales, enhancing the detection of objects of various sizes. In YOLACT, the FPN is employed to construct a multi-scale feature pyramid, enabling the network to capture object characteristics across varying levels.**ProtoNet:** ProtoNet emerges as a distinctive module within YOLACT, tasked with generating prototypes for each target instance. These prototypes embody an instance of a category and encapsulate the corresponding feature information. ProtoNet extracts features originating from different levels of the feature pyramid, employing them to generate these prototypes.**Prediction Heads:** YOLACT employs a set of prediction heads, each dedicated to predicting different target attributes, encompassing class, box position, and instance segmentation mask. These heads maintain a connection to the prototypes generated by ProtoNet, facilitating the extraction of information pertinent to target instances. Prediction heads comprise three prediction facets: mask prediction, class prediction, and box prediction. Mask prediction is primarily responsible for crafting segmentation masks for each instance. By harnessing prototypes and their associated features, the model accurately forecasts the segmentation boundaries of target instances. Class prediction determines the category to which each instance belongs, relying on the information and features drawn from the prototypes. Box prediction predicts the bounding box position for each instance, using information linked to the prototypes and features to craft precise bounding boxes.**NMS (Non-Maximum Suppression):** Following the predictions generated by the prediction heads, non-maximum suppression springs into action, sifting through the final object detection outcomes. This step eradicates redundant detections, preserving only the highest-confidence targets.**Crop Threshold Module:** The Crop Threshold module ushers in a threshold to assess the sizes of target instances. This threshold is based on the sizes of the targets and is primarily designed to distinguish between small and large objects. The crop threshold module assesses each generated segmentation mask based on a predefined threshold. If the size of the target instance is smaller than the threshold, the module crops the segmentation mask according to the predicted position of the instance. This strategic step mitigates potential errors and enhances the model’s capacity to yield precise segmentation results for smaller objects.

### 3.2. Pixel-Wise Self-Attention Module

In conventional convolutional neural networks, the resolution of feature maps gradually diminishes with increasing network depth. This limitation somewhat hinders the network’s ability to detect objects at various scales. FPN introduces a feature pyramid structure to address this constraint, enabling the network to capture multi-scale feature representations from different network levels. Consequently, FPN elevates object detection and image segmentation performance by adeptly integrating information from diverse scales. The FPN structure is portrayed in the FPN section of Figure 2.

The key concepts underpinning FPNs include the following: (1) Top-Down Pathway: FPN deploys a top-down pathway, commencing with high-level feature maps and progressively restoring the resolution of feature maps through upsampling (deconvolution). This process results in a “pyramid”-shaped feature structure, as depicted in sections C1~C5 of Figure 2. (2) Lateral Connections: Within the top-down pathway, FPN incorporates lateral connections to blend high-level feature maps with their low-level counterparts. These connections facilitate the amalgamation of detailed data from low-level feature maps with the semantic information derived from high-level feature maps, culminating in multi-scale features endowed with detail and semantics. (3) Feature Fusion: FPN fuses features sourced from the top-down pathway and lateral connections at each level. This fusion process engenders a holistic multi-scale feature representation, enabling the network to capture object characteristics across different scales more effectively. The feature maps that emerge post-fusion are illustrated as M3~M5 in Figure 2.

In the conventional FPN structure, components from both the bottom-up and top-down pathways, possessing identical resolutions, are amalgamated via straightforward linear superposition. This approach overlooks the intricate nonlinear relationships embedded within internal feature maps. We developed an innovative PSA component to redress this limitation, seamlessly integrating it into the FPN structure. The PSA component addresses this limitation by selectively capturing contextual information from varying receptive fields, thus generating more discerning features. PSA is ingeniously incorporated into the FPN structure at the red solid circles delineated in Figure 2. It conducts attention-enhancing operations on the directly superimposed feature maps to enhance features in critical regions while suppressing less important areas. The PSA component consists of two distinct attention mechanisms: channel-wise self-attention (CSA) and spatial self-attention (SSA). The operations of the CSA and SSA modules are illustrated in Figure 3.

#### 3.2.1. CSA: Channel Self-Attention Module

Each channel mapping of feature maps can be regarded as a response to specific categories, and the channels were also correlated. Drawing from [33] and using the CSA module, we harness the interdependencies among channels to enhance the specific semantic representation of features. This module is explicitly engineered to encode dependencies between channels. CSA’s specific operations are elucidated in Figure 3a.

CSA follows these steps on the original input to recalibrate the feature map through channel self-attention:

(1) Separate convolution operations are performed on the input $X\in \Omega^{C\times H\times W}$ with convolution kernels of size 1 × 1 to obtain $C_{1}\in \Omega^{C/2\times H\times W}$ and $C_{2}\in \Omega^{1\times H\times W}$, where C, H, and W represent the number of channels, height, and width of the feature maps, respectively.

(2) A Reshape operation is applied to $C_{1}$ and $C_{2}$ to obtain $C_{3}\in \Omega^{C/2\times HW}$ and $C_{4}\in \Omega^{HW\times 1\times 1}$, respectively. Simultaneously, a softmax operation is applied to $C_{4}$ to generate the feature map $C_{5}\in \Omega^{HW\times 1\times 1}$ with attention information.

(3) Element-wise multiplication is performed on $C_{3}$ and $C_{5}$, and then a convolution operation with a kernel size of 1 × 1 is applied to the multiplied result. Subsequently, the sigmoid activation function yields the channel attention feature map $C_{6}$. Multiplying the input feature map X with the attention map yields the feature map E enhanced through channel attention.

#### 3.2.2. SSA: Spatial Self-Attention Module

The SSA module is introduced to acquire dense pixel-wise contextual information essential for instance segmentation. SSA leverages all pixels within a single feature map to weigh a specific pixel. The original feature maps select location information to generate contextually relevant features, guided by the spatial attention maps. The specific operations of SSA are delineated in Figure 3b.

The SSA module performs the following steps on the original input X to recalibrate the feature map F through spatial self-attention:

(1) Separate convolution operations are applied to the input feature map X with convolution kernels of size 1 × 1 to yield the feature maps $S_{1}\in \Omega^{C/2\times H\times W}$ and $S_{2}\in \Omega^{C/2\times H\times W}$.

(2) Global pooling operations are applied to the height and width dimensions of the feature map $S_{1}$ to yield $S_{3}\in \Omega^{C/2\times 1\times 1}$. Then, a Reshape operation is performed to transform $S_{3}$, and the softmax activation function is applied to obtain the feature map $S_{6}\in \Omega^{1\times C/2}$. Additionally, for the feature map $S_{2}$, a Reshape operation is performed to obtain the feature map $S_{4}\in \Omega^{C/2\times HW}$.

(3) Element-wise multiplication is performed on $S_{4}$ and $S_{6}$ to obtain the initial spatial attention information. Subsequently, the Reshape operation and the Sigmoid activation function are employed to yield the final spatial attention map $S_{7}$. Multiplying the input X with attention $S_{7}$ results in the feature map F, enhanced through spatial attention.

The final outputs of these two attention branches are synthesized parallelly to yield the feature map Y=E+F.

## 4. Experiment

### 4.1. Experimental Parameters

The experimental setup employed six 32GB Tesla V100 GPUs, and the code was implemented using the mmdetection framework. All models were configured with a batch size of 8 and 50 iterations. Stochastic gradient descent (SGD) was employed as the optimizer, with an initial learning rate of 0.001, a momentum of 0.9, and a regularization weight decay coefficient of 0.0005. Following the default settings of mmdetection, the three-channel values of the image were normalized with a mean of (123.675, 116.28, 103.53) and a standard deviation of (58.395, 57.12, 57.375). The confidence score threshold for the test set was set to 0.05, the IOU threshold was set to 0.5, the positive sample confidence score for the training set was set to 0.5, and the negative sample confidence score was set to 0.4.

### 4.2. Evaluation Indicators

We selected average precision (AP) and average recall (AR) as the evaluation standards to evaluate the model’s performance in instance segmentation tasks for group-raised pigs. The calculation of AP involves sorting the predicted results in descending order by confidence, calculating precision and recall at various confidence thresholds and ultimately computing the area under the precision-recall curve, defining it as the average precision. Similarly, AR involves sorting the predicted results by confidence, calculating recall and precision at different confidence thresholds and determining the area under the recall-precision curve as the average recall.

We adopted the same evaluation method as COCO and three IOU thresholds of 0.5, 0.75, and 0.5–0.95:0.05 (where 0.05 represents the increment step) to measure the segmentation performance of the model under various threshold conditions, with the corresponding metrics recorded as AP_0.5_, AP_0.75_, and AP_0.5-0.95_. Additionally, considering that individual pigs may vary in size, we categorized them into small targets (individual pig pixel area < 322), medium targets (322 < individual pig pixel area < 962), and large targets (individual pig pixel area > 962). We separately calculated the AP metric for large targets, where the IOU value ranged from 0.5 to 0.95, denoting it as AP_0.5-0.95-Large_. As for the AR metric, we reported all target AR values under the IOU threshold condition of 0.5~0.95:0.05, and the AR value only for large targets denoted as AR_0.5-0.95_ and AR_0.5-0.95-Large_, respectively.

Enhancing prediction speed while maintaining accuracy is crucial in real-time instance segmentation tasks. We employed frames per second (FPS) as a speed evaluation metric, representing the number of image frames the model can process per second. A higher FPS value signifies faster inference or data processing capabilities.

### 4.3. Main Results

#### 4.3.1. Segmentation Performance of Different Backbone and Different Attention Blocks

To investigate the performance impact of various backbone networks and attention modules on pig instance segmentation tasks and select the most suitable attention module for the pig domain, we conducted two experimental settings to validate the effectiveness of the PSA module in the YOLACT real-time instance segmentation model. First, we selected ResNet50 and ResNet101 as backbone networks to assess the performance of different attention modules. Then, under identical experimental conditions, we employed three attention modules: CBAM, BAM, and SCSE, which incorporate both channel and spatial attention information, to replace the PSA module and conducted experiments on the test set. The corresponding results for the AP and AR metrics are presented in Table 2. For the CBAM module, we integrated the channel attention module (CAM) and the spatial attention module (SAM). The BAM module introduced the channel attention branch (CAB) and the spatial attention branch (SAB). The SCSE module utilized the attention mechanisms of spatial squeeze and channel excitation (cSE) and channel squeeze and spatial excitation (sSE).

**Impact of different backbone networks:** ResNet50 and ResNet101 are variants of the ResNet architecture, comprising stacked residual blocks and a global average pooling layer. The distinction lies in the model’s depth and the number of parameters. We conducted comparative experiments under controlled conditions to assess the impact of using different backbone networks on the YOLACT model’s performance. Without incorporating any attention information, utilizing ResNet101 instead of ResNet50 yielded improved results. With ResNet101, it achieved AP series metrics of 0.892, 0.718, 0.610, and 0.629 for AP_0.5_, AP_0.75_, AP_0.5-0.95_, and AP_0.5-0.95-Large_, respectively. This represented enhancements of 1.4%, 2.5%, 2.3%, and 2.3%, respectively, compared to using ResNet50 as the backbone network. For the AR metric, ResNet101 also outperformed ResNet50, achieving AR_0.5-0.95_ and AR_0.5-0.95-Large_ values of 0.678 and 0.697, respectively, which were 2.2% higher than those attained by ResNet50 with corresponding indicators. Deeper models, such as ResNet101, often capture more features and patterns, potentially explaining its superior performance in complex tasks. However, the increased number of layers and parameters in ResNet101 results in a larger model size and complexity, necessitating more computational resources and time during training and inference. Notably, adopting ResNet101 led to a notable decrease in the FPS value, from 35.51 to 28.39. Therefore, careful consideration and balance are required in practical production scenarios when selecting the backbone network.

**Impact of different attention modules:** For the same backbone network, adding different attention modules introduced a certain degree of performance variation. Nevertheless, our proposed PSA attention module consistently delivered the best results. Specifically, CBAM, BAM, and SCSE, which incorporated both channel and spatial attention information, exhibited comparable performance, with BAM and SCSE generally outperforming CBAM. When ResNet101 served as the backbone network, BAM achieved superior AP metrics, reaching 0.892, 0.733, 0.611, and 0.641 for AP_0.5_, AP_0.75_, AP_0.5-0.95_, and AP_0.5-0.95-Large_, respectively. These metrics were 0.5%, 0.1%, 0.1%, and 1.1% higher than BAM. CBAM enhanced feature information by sequentially applying channel and spatial attention, first learning channel attention and then spatial attention in the spatial dimension. While this allowed it to focus better on features from different channels and positions, it introduced increased dependencies among hierarchical features. Errors in obtaining channel attention content could affect the acquisition of spatial attention information through gradient propagation. In contrast, BAM combined channel and spatial attention content in parallel, avoiding the cascading errors inherent in the sequential approach. This contributed to better model performance. Additionally, models utilizing BAM and SCSE attention modules achieved higher FPS values than CBAM, primarily because CBAM introduced global average pooling and fully connected operations, which imposed significant computational overhead when processing feature maps. Despite promising results from BAM and SCSE, our PSA module further enhanced model performance. When ResNet50 served as the backbone network, PSA outperformed the SCSE attention module in terms of AP_0.5_, AP_0.75_, AP_0.5-0.95_, and AP_0.5-0.95-Large_, AR_0.5-0.95_, and AR_0.5-0.95-Large_ metrics, achieving improvements of 1.8%, 2.1%, 1.3%, 1.2%, 0.9%, and 1.4%, respectively. PSA’s ability to introduce attention mechanisms at different scales strengthened the representation of multi-scale features. This allowed the model to simultaneously focus on fine-grained details and global contextual information in the image, thereby enhancing the expressiveness of its features. Therefore, the proposed PSA module can enhance segmentation accuracy while maintaining segmentation speed in practical production settings.

#### 4.3.2. Detection Performance of Different Backbone and Different Attention Blocks

In the context of instance segmentation, which includes an object detection branch, we conducted experiments utilizing ResNet50 and ResNet101 as backbone networks. We compared the results with no attention, BAM, CBAM, and SCSE attention mechanisms. The experimental results, showcasing various evaluation metrics, are presented in Table 3. The FPS metric, which is identical for both object detection and instance segmentation, has been omitted from Table 3. However, the selection of other metrics aligns consistently with Table 2.

**Impact of different conditions:** The data in Table 3 indicate two significant trends. First, introducing any attention module leads to performance improvements compared with no attention module, irrespective of the selected backbone network. For example, in the ResNet50 backbone with the BAM attention module, the AP_0.5_ and AP_0.75_ metrics exhibited improvements of 1.1% and 0.7%, respectively, over the no-attention baseline (NONE). Similarly, for the CBAM module, compared to the NONE baseline, the AP_0.5-0.95-Large_ and AR_0.5-0.95_ metrics improved by 2.0% and 1.8%, respectively, indicating the effectiveness of employing attention mechanisms for performance enhancement. Second, the proposed PSA module consistently performs better than other attention modules. For instance, with the ResNet101 backbone, the PSA module outperforms the CBAM module by 1.6% in the AP_0.5-0.95-Large_ and AR_0.5-0.95_ metrics, highlighting the superiority of the proposed attention mechanism. Compared with existing attention modules, PSA is better suited for the instance segmentation of pigs. Furthermore, ResNet101 outperforms ResNet50 as the backbone network under identical conditions, primarily due to its deeper network architecture, which captures more comprehensive image features.

### 4.4. Visualization of Prototype Mask

During the object detection and segmentation stages, the YOLACT model does not accurately predict instance segmentation masks for target objects. Instead, it employs Prototype maps to generate the segmentation results. Specifically, the model assigns a Prototype map for each detected target instance and adjusts its scale and position based on the instance’s location and size in the original image. This approach significantly reduces the number of parameters required for prediction, thus enhancing its real-time performance. With ResNet101 as the backbone network, the Prototype maps of the YOLACT model with the incorporated PSA attention module are illustrated in Figure 4. We selected two images from the test set for analysis. The results in the fourth column are obtained by element-wise multiplication of the corresponding prototype mask and mask coefficient in the second and third columns, respectively. As illustrated in Figure 4, for specific individual pigs, the mask coefficient assigns negative scores to areas containing background information or unrelated individuals, attenuating the influence of these irrelevant regions on the segmentation prediction for the current pig individual.

### 4.5. Visualization of Attention Information

To visually demonstrate the effectiveness of the attention mechanism in the instance segmentation task for pigs, we employed ResNet101 as the backbone network and visualized attention maps after integrating the PSA attention content. The specific results are shown in Figure 5, with brightness corresponding to the activation value. Brighter areas indicated larger activation values. We considered two different shooting perspectives: a horizontal view (first and second rows of results) and a top-down view (third and fourth rows of results), as well as daytime (first, second, and fourth rows) and nighttime (third row) scenarios. By visualizing the attention maps in these different scenarios, we can assess how the attention mechanism adapts to various viewing angles and lighting conditions, evaluating its effectiveness in guiding the segmentation process across diverse environmental settings.

Figure 5 shows that different attention mechanisms focus on distinct regions. Brighter areas in the second column appear more oriented towards the individual pig, while the fourth column emphasizes background information. Combining these two attention mechanisms allows for a more effective separation of the pig from complex scenes. Additionally, the fifth column emphasizes the pig’s edge and contour information, contributing to a more accurate decision boundary between the pig and the background. Consequently, with the integration of the PSA attention module, reasonable attention knowledge can be extracted to a certain extent, demonstrating the effectiveness of this module.

### 4.6. Visualization of Prediction Results for Different Scenarios

We selected CBAM, BAM, and SCSE as comparative attention modules to visually illustrate the predictive results across various scenarios. We present visualization results from two data collection perspectives: a horizontal view and a top-down view, considering both nighttime and daytime data collection times to represent distinct scenarios. The visualization of predicted outcomes is depicted in Figure 6, where the regions enclosed by red circles in the first column indicate locations with significant differences in predictive outcomes among different attention mechanisms.

First, our PSA module consistently delivers accurate predictions in cases involving individual pigs situated within a localized region visible in the video frame, as exemplified by the pig labeled as ④. In contrast, alternative attention modules falter in forecasting the presence of the pig in such instances. Second, in regions identified as ③, the pig predicted by the PSA module exhibits a more comprehensive and distinct outline. This module accurately distinguishes between the pig and background content, even when obstructions such as pigpens occlude the view. Lastly, in nighttime scenarios, despite this portion of data not being part of the model’s training regimen, the PSA module consistently generates impressive predictions. This phenomenon is particularly pronounced where the distinction between individual pigs and their surroundings is challenging for the human eye to discern, as exemplified by the region marked as ⑦. These visualizations underscore the PSA module’s robustness across diverse viewpoints and scenarios.

## 5. Discussion

Despite the impressive performance exhibited by the PSA model, outperforming existing attention modules such as BAM, CBAM, and SCSE, two notable shortcomings warrant further investigation:The integration of the PSA module typically necessitates the computation of weights or attention scores for various positions within the input. This process may introduce additional computational complexity to the model, resulting in longer training and inference times.The successful implementation of the PSA module often relies on a substantial volume of training data to accurately learn effective attention distributions. If the training data are insufficient or do not adequately represent all potential scenarios, the attention mechanism may struggle to generalize effectively to new data. Therefore, future work will prioritize expanding the dataset and enhancing the quality of dataset annotations.Additionally, our focus has shifted towards the refinement of instance segmentation algorithms. This includes optimizing model segmentation speed and improving segmentation performance. However, the practical deployment of the model in real-world production environments requires further investigation. For example, we must explore strategies for migrating the model to mobile applications, enabling its effective use in real-world production settings.

## 6. Conclusions

We introduced a plug-and-play PSA attention module that incorporates channel and spatial attention information for multi-faceted feature selection from both channel and spatial perspectives. To validate the effectiveness of PSA, we embedded it into the YOLACT model, conducting experiments using ResNet50 and ResNet101 as backbone networks, and derived the following four conclusions:

(1) We comprehensively compared the performance of the PSA module with existing BAM, CBAM, and SCSE attention modules in instance segmentation and object detection. This analysis confirmed that the PSA module has stronger feature extraction capabilities and achieves superior performance.

(2) By visualizing the prototype vectors in the YOLACT model, we provided a more intuitive perspective on the interpretability of the model’s predictions after incorporating PSA attention.

(3) Visualizing the attention feature maps generated by the PSA attention module provided a direct view of the attention knowledge that PSA can capture, indirectly confirming the necessity of introducing the PSA module.

(4) We verified the adaptability and robustness of the YOLACT model with the PSA attention module in different viewpoints and nighttime scenes, confirming that the model exhibits a degree of transferability.

The study of pig instance segmentation has important agricultural and economic significance. By accurately identifying and segmenting individuals in pig images, researchers can effectively monitor pig health, behavioral characteristics, and growth and development. This technology not only helps improve the production efficiency of animal husbandry, optimize feeding management, and reduce losses during the breeding process, but also provides key technical support for precision breeding and intelligent breeding, and promotes the sustainable development of the agricultural industry.

While incorporating the PSA module led to performance improvements, there are two limitations to consider:

(1) Introducing the PSA module adds a new plug-and-play component to the original network structure, which inevitably increases the number of parameters. This may have some impact on the training and inference speed of the model, although this impact is considered acceptable.

(2) The transferability performance of the PSA module needs further validation in more complex environments. However, due to limitations in the scale of the current dataset and the covered application scenarios, it is possible that some scenarios were not adequately covered. In future research, we will focus on addressing this issue.

## Figures and Tables

**Figure 1 animals-13-03591-f001:**
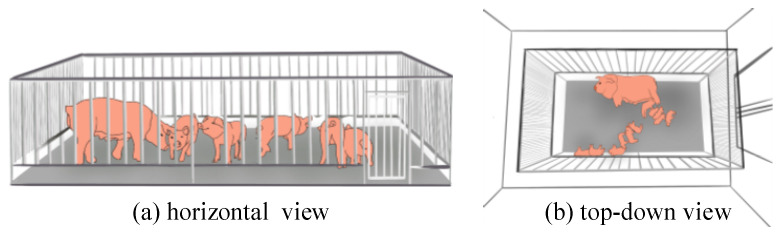
Data collection diagram.

**Figure 2 animals-13-03591-f002:**
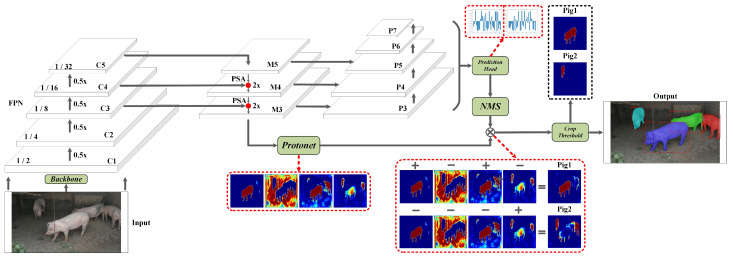
YOLACT model structure diagram.

**Figure 3 animals-13-03591-f003:**
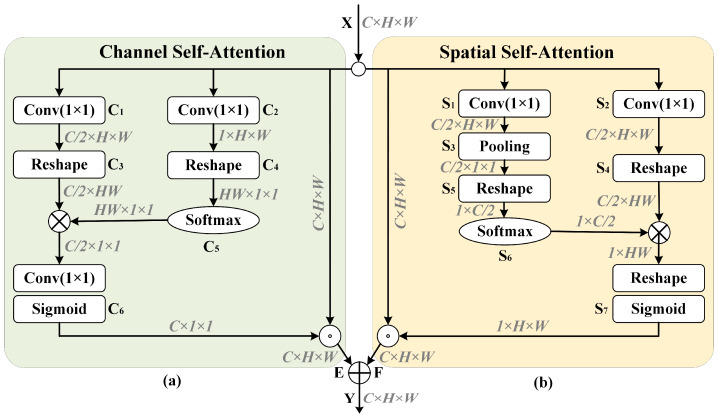
PSA attention module structure diagram. (**a**) represents channel self-attention module, (**b**) represents spatial self-attention module.

**Figure 4 animals-13-03591-f004:**
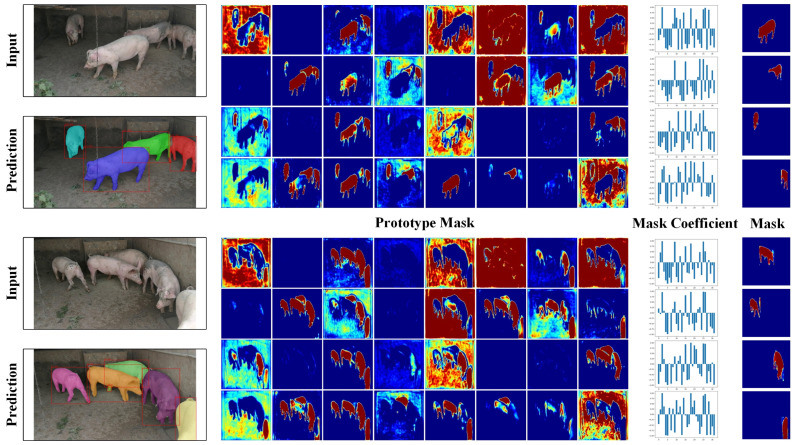
Visualization of the prototype diagram; the result of the fourth column is obtained by multiplying the second column and the third column.

**Figure 5 animals-13-03591-f005:**
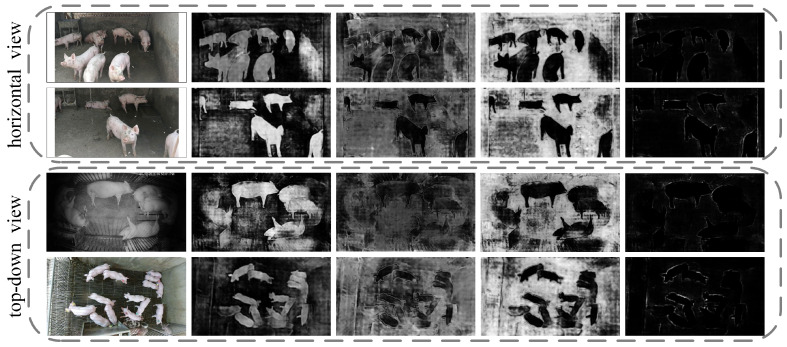
Selective visualization of attention information.

**Figure 6 animals-13-03591-f006:**
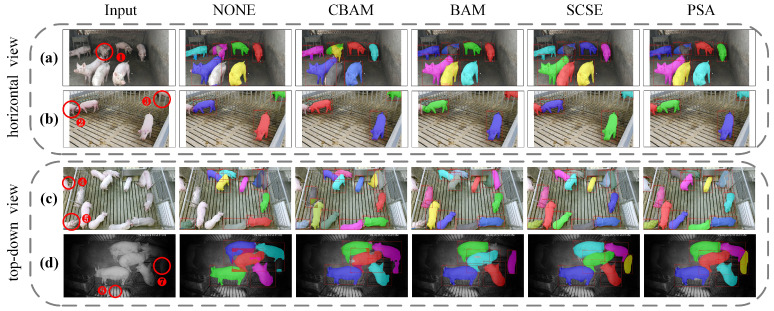
Visualization of prediction results by different attention mechanisms in various scenarios. (**a**) and (**b**) respectively represent two examples of horizontal viewing angles, and (**c**) and (**d**) respectively represent two examples of top-down vertical viewing angles. ①–⑦ indicate the parts that need to be focused on.

**Table 1 animals-13-03591-t001:** Data augmentation operations.

Augmentation Method	Parameter Settings	Probability
Translation zoom and rotate	The translation factor is 0.0625, the image scaling and rotation factors are set to 0.1~0.3, and linear interpolation is used to fill the area where the translation occurs.	0.5
Randomly change brightness and contrast	The brightness and contrast variation range factors are both set to 0.1~0.3.	0.2
RGB value transformation	The R/G/B three-channel random transformation range is set to 0~10.	0.1
HSV value transformation	The range of H/S/V random transformation is set to 0~20, 0~30, and 0~20, respectively.	0.1
Image compression	The upper and lower limits of the compression percentage are set to 95 and 85, respectively.	0.2
Randomly rearrange channels	——	0.1
Median blur	The filter radius is set to 3.	0.1

**Table 2 animals-13-03591-t002:** The segmentation performance of AP and AR index values of different attention modules under the condition of different backbone networks.

Backbone	Attention Block	FPS	AP_0.5_	AP_0.75_	AP_0.5-0.95_	AP_0.5-0.95-Large_	AR_0.5-0.95_	AR_0.5-0.95-Large_
ResNet50	NONE	35.5	87.8	69.3	58.7	60.6	65.6	67.5
	BAM	31.1	88.7	71.0	60.0	61.8	67.6	69.5
	CBAM	30.2	87.8	70.8	59.6	61.4	66.4	68.4
	SCSE	31.3	88.3	72.0	60.7	62.7	67.3	69.4
	PSA	33.6	**90.1**	**74.1**	**62.0**	**63.9**	**68.2**	**70.8**
ResNet101	NONE	28.4	89.2	71.8	61.0	62.9	67.8	69.7
	BAM	26.5	89.2	73.3	61.4	64.1	68.8	70.8
	CBAM	25.3	88.7	73.2	61.3	63.0	68.1	69.8
	SCSE	25.8	89.2	73.6	61.6	63.6	68.9	70.9
	PSA	27.2	**91.5**	**75.1**	**63.7**	**65.4**	**70.2**	**72.1**

Note that bold sections indicate the best performing results.

**Table 3 animals-13-03591-t003:** The detection performance of AP and AR index values of different attention modules under the condition of different backbone networks.

Backbone	Attention Block	AP_0.5_	AP_0.75_	AP_0.5-0.95_	AP_0.5-0.95-Large_	AR_0.5-0.95_	AR_0.5-0.95-Large_
ResNet50	NONE	90.1	73.3	60.5	62.4	66.4	68.4
	BAM	91.2	74.0	62.0	63.8	67.9	69.6
	CBAM	91.0	73.2	62.5	64.4	68.2	70.2
	SCSE	91.2	73.3	61.4	63.2	67.3	69.2
	PSA	**92.1**	**75.4**	**63.0**	**64.8**	**68.9**	**70.9**
ResNet101	NONE	91.1	74.5	62.7	64.6	68.3	70.3
	BAM	**92.1**	75.3	63.3	65.2	69.5	71.5
	CBAM	91.1	76.2	63.1	64.9	69.1	70.8
	SCSE	91.1	74.2	63.4	65.1	69.2	71.2
	PSA	91.9	**78.0**	**64.8**	**66.7**	**70.7**	**72.8**

Note that bold sections indicate the best performing results.

## Data Availability

Data included in the text.

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
