# Peer review of "Pixel Self-Attention Guided Real-Time Instance Segmentation for Group Raised Pigs"

_animals, 2023, doi:10.3390/ani13233591_

Round 1
Reviewer 1 Report
Comments and Suggestions for Authors
Review comments on “Pixel Self-Attention Guided Real-time Instance Segmentation for Group Raised Pigs” by Zongwei Jia etl.
This work presents a novel pixel self-attention (PSA) module by combining channel attention and spatial attention. This module was integrated into the feature pyramid module of the YOLACT model.
My main general comments are as below:
- The authors didn’t provide a comparison of the performances on training and testing sets. The authors should investigate experimentally the overfitting of the proposed module.
- The authors should share the dataset to be downloaded freely to the industry and research community.
- The work will be significant if the source codes are presented to the public for a detailed analysis of the proposed method and model.
- Conclusions need more elaboration about: outcomes, limitations, and possible/future scenarios.
- The authors should investigate the stability of the proposed model because image can be degraded by additive noise, in the presence of cluttering backgrounds, geometric modifications such as pose changing and scaling, nonuniform illumination, and eventual object occlusions.
Author Response
Reply to Reviewer
Thanks for your time and efforts devoted to the review of our paper, which are much appreciated. All your comments have been addressed in this response.
General comments:
Comment: “This work presents a novel pixel self-attention (PSA) module by combining channel attention and spatial attention. This module was integrated into the feature pyramid module of the YOLACT model.”
Response: Thanks so much indeed for your positive comments.
Specific comments:
Comment: “The authors didn’t provide a comparison of the performances on training and testing sets. The authors should investigate experimentally the overfitting of the proposed module.”
Response: Thanks for your insightful comment. The main reason for not providing numerical results for the training set predictions is that we followed the common practice in the existing literature, which only reports predictive performance on the test set. In the context of general dataset partitioning, the training set is primarily used for model training, the validation set is primarily used for tuning hyperparameters, and the test set is mainly used to assess the generalization performance of the model trained on the training set. As you mentioned, presenting results for the training set can serve as an indirect indicator of whether the model has overfit. To illustrate this, we provide the performance comparison results for backbone networks ResNet50 and ResNet101 on both the training and test sets, as shown in Table 1:
|
Backbone |
Attention Block |
AP0.5 |
AP0.75 |
AP0.5-0.95 |
AP0.5-0.95-Large |
AR0.5-0.95 |
AR0.5-0.95-Large |
|
ResNet50 |
PSA-Train |
0.945 |
0.786 |
0.679 |
0.700 |
0.712 |
0.776 |
|
PSA-Test |
0.901 |
0.741 |
0.620 |
0.639 |
0.682 |
0.708 |
|
|
ResNet101 |
PSA-Train |
0.967 |
0.799 |
0.703 |
0.721 |
0.742 |
0.802 |
|
PSA-Test |
0.915 |
0.751 |
0.637 |
0.654 |
0.702 |
0.721 |
Table 1: Performance comparison on training set and test set
We can observe that the performance on the training set is indeed slightly higher than that on the test set, but there isn't a drastic difference. The fact that there is still a gap between the performance on the training set and 100% also indirectly suggests that our dataset presents some challenges.
Comment: “The authors should share the dataset to be downloaded freely to the industry and research community.”
Response: Thanks for your insightful comment. We haven't open-sourced the dataset temporarily mainly for the following reasons:
- Firstly, the dataset is still continuously expanding, including different scenarios such as daytime and nighttime, different age groups of individual pigs, and datasets captured from different perspectives. We intend to release a large-scale dataset related to pig research after completing comprehensive data collection, which is a long-term task.
- Secondly, regarding dataset labeling, we are currently working on real-time instance segmentation tasks. In addition to this, we are also creating datasets for pig keypoint detection, pig object detection, and pig behavior tracking. These datasets share the same data sources.
- Finally, we plan to write a dedicated dataset paper related to pig research and open-source all our baseline code and the complete dataset. However, I believe you may need to wait for a while. If you are interested in this dataset, please feel free to contact us via email at any time.
Comment: “The work will be significant if the source codes are presented to the public for a detailed analysis of the proposed method and model.”
Response: Thanks for your insightful comment. Our code is implemented based on the open-source framework mmdetection[1]. Your suggestions are highly valuable, and we will prioritize open-sourcing the code after refactoring, just as we have done with our previous work[2]. We will publish the code and the entire code execution process on GitHub. Thank you for your valuable feedback.
Comment: “Conclusions need more elaboration about: outcomes, limitations, and possible/future scenarios.”
Response: Thanks for your insightful comment. We have rewritten the conclusion part of the original manuscript. The rewritten content is as follows:
In this study, we introduced a plug-and-play PSA attention module that incorporates channel and spatial attention information for multi-faceted feature selection from both channel and spatial perspectives. To validate the effectiveness of PSA, we embedded it into the YOLACT model, conducting experiments using ResNet50 and ResNet101 as backbone networks, and derived the following four conclusions:
- We comprehensively compared the performance of the PSA module with existing BAM, CBAM, and SCSE attention modules in instance segmentation and object detection. This analysis confirmed that the PSA module has stronger feature extraction capabilities and achieves superior performance.
- By visualizing the prototype vectors in the YOLACT model, we provided a more intuitive perspective on the interpretability of the model's predictions after incorporating PSA attention.
- Visualizing the attention feature maps generated by the PSA attention module provided a direct view of the attention knowledge that PSA can capture, indirectly confirming the necessity of introducing the PSA module.
- We verified the adaptability and robustness of the YOLACT model with the PSA attention module in different viewpoints and nighttime scenes, confirming that the model exhibits a degree of transferability.
While incorporating the PSA module led to performance improvements, there are two limitations to consider:
- Introducing the PSA module adds a new plug-and-play component to the original network structure, which inevitably increases the number of parameters. This may have some impact on the training and inference speed of the model, although this impact is considered acceptable.
- The transferability performance of the PSA module needs further validation in more complex environments. However, due to limitations in the scale of the current dataset and the covered application scenarios, it is possible that some scenarios were not adequately covered. In future research, we will focus on addressing this issue.
Comment: “The authors should investigate the stability of the proposed model because image can be degraded by additive noise, in the presence of cluttering backgrounds, geometric modifications such as pose changing and scaling, nonuniform illumination, and eventual object occlusions.”
Response: Thanks for your insightful comment. Your advice is highly valuable, and I believe that the stability of the model is indeed a topic worthy of research. I think in terms of model stability, we do have some content in the original manuscript. For example, in Section 4.6, we presented the model's predictive performance in different scenarios to validate its transferability. As mentioned in the manuscript, for some of these scenario data, we obtained them directly from third-party datasets, rather than collecting them ourselves. Since the third-party datasets did not include instance segmentation annotations, we were unable to provide corresponding numerical results and could only showcase them through visual predictions. In our future work, we will focus on instance segmentation of group-housed pigs under complex conditions, which I believe is a topic worth delving into. Your suggestions have provided us with significant inspiration.
[1] https://github.com/open-mmlab/mmdetection
[2] https://github.com/zhiweihu1103/pig-instance-segmentation
Reviewer 2 Report
Comments and Suggestions for Authors
General Comments
This is a very interesting manuscript addressing an important issue in animal science, namely, the identification of animals in a production system. Once perfected, this would be a useful tool for producers.
Specific Comments:
1. The authors suggest the Pixel Self Attention (PSA) algorithm can assist producers in the identification of animals with aberrations in health and behaviour (example Discussion L553). However, no animal health or behaviour models are provided for this proof. It would therefore be suggested that the authors modify their claims by suggesting their PSA model may be useful for this purpose subject to further testing.
2. The authors discuss the threshold segmentation approach for their data. How were the thresholds established? For example ROC curves for optimal thresholds?
3. For the segmentation models, how were image boundary and edge conditions established?
4. The authors suggest animal clustering algorithms were established. Were these segmentation algorithms novel for their work or did the authors use established segmentation algorithms?
5. The three animal size models chosen are quite different. How does the PSA model compare for example on littermates or pen mates of similar size/weight.
6. It is curious for all the models utilized by the authors no mention or comparison to industry standard practice models such as tensorflow were used or compared?
7. The figures demonstrate pseudo colours for identified animals. Is there a numerical limit to the number of animals that can be identified within a pen?
8. Again, the authors discuss the usefulness of their PSA technique but provide no testable behaviours, health aberrations or metabolic differences to demonstrate their model effectiveness. Applying their PSA model against or with known behaviour, health or metabolic events would be useful and ultimately probably necessary to be utilized in production facilities.
9. For the image analysis, how were the horizon and boundary effects managed.
10. The authors report that for optimal function the PSA model requires the use of animal weight predictions. How were these predictions obtained? Were gravimetric or image size systems used?
11. It would be useful if the authors could contrast and compare their PSA model against known identification systems such as low and high frequency RFIDs.
12. L476 suggests improvements of 1-2%. Is this practically meaningful? Were any of the test performance parameters compared using conventional test evaluation such as with Yoden index?
Author Response
Reply to Reviewer
Thanks for your time and efforts devoted to the review of our paper, which are much appreciated. All your comments have been addressed in this response.
General comments:
Comment: “This is a very interesting manuscript addressing an important issue in animal science, namely, the identification of animals in a production system. Once perfected, this would be a useful tool for producers.”
Response: Thanks so much indeed for your positive comments.
Specific comments:
Comment: “The authors suggest the Pixel Self Attention (PSA) algorithm can assist producers in the identification of animals with aberrations in health and behaviour (example Discussion L553). However, no animal health or behaviour models are provided for this proof. It would therefore be suggested that the authors modify their claims by suggesting their PSA model may be useful for this purpose subject to further testing.”
Response: Thanks for your insightful comment. This suggestion is very insightful. After careful consideration, we have decided to remove the mentioned content raised by the reviewer, such as the feasibility of this model in the context of pig disease prevention. We acknowledge that, at present, we lack experimental data to support this claim. We appreciate the reviewer's feedback. We have rewritten the conclusion part of the original manuscript. The rewritten content is as follows:
In this study, we introduced a plug-and-play PSA attention module that incorporates channel and spatial attention information for multi-faceted feature selection from both channel and spatial perspectives. To validate the effectiveness of PSA, we embedded it into the YOLACT model, conducting experiments using ResNet50 and ResNet101 as backbone networks, and derived the following four conclusions:
- We comprehensively compared the performance of the PSA module with existing BAM, CBAM, and SCSE attention modules in instance segmentation and object detection. This analysis confirmed that the PSA module has stronger feature extraction capabilities and achieves superior performance.
- By visualizing the prototype vectors in the YOLACT model, we provided a more intuitive perspective on the interpretability of the model's predictions after incorporating PSA attention.
- Visualizing the attention feature maps generated by the PSA attention module provided a direct view of the attention knowledge that PSA can capture, indirectly confirming the necessity of introducing the PSA module.
- We verified the adaptability and robustness of the YOLACT model with the PSA attention module in different viewpoints and nighttime scenes, confirming that the model exhibits a degree of transferability.
While incorporating the PSA module led to performance improvements, there are two limitations to consider:
- Introducing the PSA module adds a new plug-and-play component to the original network structure, which inevitably increases the number of parameters. This may have some impact on the training and inference speed of the model, although this impact is considered acceptable.
- The transferability performance of the PSA module needs further validation in more complex environments. However, due to limitations in the scale of the current dataset and the covered application scenarios, it is possible that some scenarios were not adequately covered. In future research, we will focus on addressing this issue.
Comment: “The authors discuss the threshold segmentation approach for their data. How were the thresholds established? For example ROC curves for optimal thresholds?”
Response: Thanks for your insightful comment. I think what you are referring to is the content in Table 1. In Table 1, there are two types of values mentioned. One corresponds to the thresholds involved in data augmentation operations (Table's 2nd column). These are empirical parameters. The other refers to the probabilities associated with the occurrence of data augmentation operations (Table's 3rd column). These probability values are used to bias certain data augmentation operations. However, neither of these numerical values produces ROC curves, so we may not be able to provide corresponding results. Nevertheless, the reviewing has proposed a good idea. When designing the thresholds, we rely on existing empirical knowledge. In the future, we need to work on automating the selection of these thresholds.
Comment: “For the segmentation models, how were image boundary and edge conditions established?”
Response: Thanks for your insightful comment. As you said, in image instance segmentation task, establishing image boundaries and edge conditions is crucial. It helps define the model's predictions, ensuring that the segmentation results exhibit good accuracy and continuity near the boundaries. Common methods include edge detection and boundary-based loss functions. For edge detection, techniques like Canny edge detection, Sobel operator, Scharr operator, and others are typically employed to select boundaries. However, it's worth noting that this approach heavily relies on the dataset. In the case of boundary-based loss functions, they use loss metrics such as cross-entropy loss or Dice coefficient to penalize inconsistencies between the segmentation results and the image edges. Considering that YOLACT itself embeds the cross-entropy loss function within its framework, and since our model is developed based on YOLACT, so we've implemented edge selection through a boundary-based loss function.
Comment: “The authors suggest animal clustering algorithms were established. Were these segmentation algorithms novel for their work or did the authors use established segmentation algorithms?”
Response: Thanks for your insightful comment. No, we did not use animal clustering algorithms. The method in this paper is based on convolutional neural networks and deep learning techniques. There is no intersection with clustering algorithms. However, clustering algorithms can indeed be used in the field of animal segmentation, although we have not attempted to do so at this time.
Comment: “The three animal size models chosen are quite different. How does the PSA model compare for example on littermates or pen mates of similar size/weight.”
Response: Thanks for your insightful comment. Typically, for the sake of model transferability and robustness, we consider various sizes of pigs as our subjects of study. The goal is to enable the model to learn richer features from a more diverse range of scenes, rather than being limited to a specific pigsty or a particular pig size. Therefore, in the data collection part of the entire article, we integrated data from different age groups, pigsties, and varying light conditions to create a comprehensive dataset. Models trained on this dataset are expected to exhibit strong generalization performance. Your advice is highly valuable, and it provides a constructive direction for us to further test the model's transferability.
Comment: “It is curious for all the models utilized by the authors no mention or comparison to industry standard practice models such as tensorflow were used or compared?”
Response: Thanks for your insightful comment. It's important to clarify a few points. Firstly, this article is primarily focused on the field of instance segmentation, which is a relatively new research direction that emerged after the advent of deep learning techniques. The reviewer's mention of tensor flow and related topics appears to pertain more to semantic segmentation. Secondly, the methods we've selected are predominantly based on deep learning techniques, and we haven't considered traditional segmentation algorithms. Lastly, our primary objective is to validate that our proposed PSA module is more effective than other attention modules. Therefore, direct comparisons with traditional segmentation algorithms will be de-emphasized.
Comment: “The figures demonstrate pseudo colours for identified animals. Is there a numerical limit to the number of animals that can be identified within a pen?”
Response: Thanks for your insightful comment. There is no limit on the number of animals in a pen, and our model can segment images with a variable number of group-housed pigs.
Comment: “Again, the authors discuss the usefulness of their PSA technique but provide no testable behaviours, health aberrations or metabolic differences to demonstrate their model effectiveness. Applying their PSA model against or with known behaviour, health or metabolic events would be useful and ultimately probably necessary to be utilized in production facilities.”
Response: Thanks for your insightful comment. We have deleted similar descriptions in the article. There are indeed some problems with this. Thank you to the reviewer for correcting it.
Comment: “For the image analysis, how were the horizon and boundary effects managed.”
Response: Thanks for your insightful comment. As you said, managing horizon and boundary effects in image analysis is crucial to ensure accurate and consistent results. These effects can introduce artifacts and distortions in the analysis, especially when processing images that have objects or features near their edges. In fact, we introduced some ways in the data augmentation part to be able to do the horizon and boundary effects managed, such as translation zoom and rotate. In addition, YOLACT is implemented based on convolutional neural network technology, and the convolution kernel in the convolutional neural network is very effective for the horizon and boundary effects managed, because there is a padding operation during the entire convolution operation. The padding can be done using various methods, such as zero-padding (adding zeros), edge-padding (repeating the edge values), or mirror-padding (mirroring the existing values). Padding provides a buffer zone that allows for more accurate analysis without being affected by boundary artifacts. Therefore, YOLACT can learn well about boundary information, so YOLACT after adding the PSA module also has this ability.
Comment: “The authors report that for optimal function the PSA model requires the use of animal weight predictions. How were these predictions obtained? Were gravimetric or image size systems used?”
Response: Thanks for your insightful comment. I think what you should be talking about is this sentence in the 4.1. Experimental parameters section: "a regularization weight decay coefficient of 0.0005". I need to explain this. The weight here does not mean body weight. This is the hyperparameter in the optimization function. Our article does not involve function calculations related to pig weight.
Comment: “It would be useful if the authors could contrast and compare their PSA model against known identification systems such as low and high frequency RFIDs.”
Response: Thanks for your insightful comment. This is a very valuable suggestion. In fact, we are currently conducting some related research, but it goes beyond the scope of this paper. Our main focus in this article is to demonstrate the effectiveness of the attention mechanism we designed for instance segmentation in group-housed pigs. The animal identification system using RFID technology, which does not involve deep learning or attention mechanisms, was not considered in this paper. However, your insightful suggestion provides an inspiring research topic, and we will explore it further in our future studies.
Comment: “L476 suggests improvements of 1-2%. Is this practically meaningful? Were any of the test performance parameters compared using conventional test evaluation such as with Yoden index?”
Response: Thanks for your insightful comment. Firstly, a 1-2% improvement is meaningful because we can observe that YOLACT's accuracy is already relatively high. To further enhance the model's performance on top of this high accuracy, a 1-2% improvement is acceptable. Secondly, the enhancements achieved by the baseline BAM, CBAM, and SCSE, relative to YOLACT without attention mechanisms, are also limited. Therefore, the limitations in improvement are a common phenomenon across models using attention mechanisms. Lastly, considering that the baseline BAM, CBAM, and SCSE models did not involve Yoden index calculations, we followed a standardized paradigm and did not compute related values. However, in future research, we will consider incorporating metrics related to the Yoden index.
Reviewer 3 Report
Comments and Suggestions for Authors
In the summary and abstract suggest having the same points about why and how this technology could help industry
You mention how many pigs and the age of the pigs from which data is collected in the abstract but I am missing that key information in the methods
The second sentence of the intro does not align with common industry terminology for group housed pigs. Reword this to align with common terminology.
You introduction and summary/abstract mentions breeding of pigs yet you used data from growing pigs, not of pigs that are near breeding age or intended for that purpose. Restructure your argument to remove this point as there are too many unknowns in applying your model to such pigs at this time.
As I understand it you are using data from someone else's study that used a different ML algo to process the images? How exactly is your data collection more novel by using a different viewpoint if someone else did very similar work? Where is all the data about the pigs and the age/weight of these pigs?
Strongly suggest staying in third person throughout the paper, revise accordingly throughout
Line 233 grammar error please correct
Should the equations on page 8 be in the journal equation form and not imbedded with the text?
varying sig figs in the results, standardize throughout.
What is the "so what" of this work? a better way for the computer to recognize a pig in a video? seems like the point that it will help managers id sick pigs and manage without a stronger lead on to how this work will help the industry
Why is there a "(a)" after the Figure call out in text?
Comments on the Quality of English Languagevary person throughout the paper
Author Response
Reply to Reviewer
Thanks for your time and efforts devoted to the review of our paper, which are much appreciated. All your comments have been addressed in this response.
Specific comments:
Comment: “In the summary and abstract suggest having the same points about why and how this technology could help industry.”
Response: Thanks for your insightful comment. We have removed inconsistent descriptions.
Comment: “You mention how many pigs and the age of the pigs from which data is collected in the abstract but I am missing that key information in the methods.”
Response: Thanks for your insightful comment. We added a description in 2.1. Data Collection.
Comment: “The second sentence of the intro does not align with common industry terminology for group housed pigs. Reword this to align with common terminology.”
Response: Thanks for your insightful comment. I have modified.
Comment: “You introduction and summary/abstract mentions breeding of pigs yet you used data from growing pigs, not of pigs that are near breeding age or intended for that purpose. Restructure your argument to remove this point as there are too many unknowns in applying your model to such pigs at this time.”
Response: Thanks for your insightful comment. I've removed the corresponding description.
Comment: “As I understand it you are using data from someone else's study that used a different ML algo to process the images? How exactly is your data collection more novel by using a different viewpoint if someone else did very similar work? Where is all the data about the pigs and the age/weight of these pigs?.”
Response: Thanks for your insightful comment. First of all, our dataset consists of two parts. One part of the data set is collected by ourselves and is mainly used for model training and testing. The other part is data from a third party and is mainly used for testing the migration ability of the model. Secondly, the most significant distinction between our dataset and existing datasets is the data collection method. We collected our data using horizontal view, whereas the existing datasets were obtained through top-down view. Additionally, our dataset includes individual pigs at different stages of adhesion, which is not covered in the existing datasets. Finally, we integrated the datasets from different adhesion stages of pigs for training without separately counting the number of images for each adhesion stage. The purpose behind this approach is to ensure that the model receives a more diverse set of inputs, without constraining the image proportions for each adhesion stage. This allows the trained model to exhibit better robustness. Additionally, we plan to open-source our data in the future.
Comment: “Strongly suggest staying in third person throughout the paper, revise accordingly throughout.”
Response: Thanks for your insightful comment. I have modified the corresponding content.
Comment: “Line 233 grammar error please correct.”
Response: Thanks for your insightful comment. I have modified the corresponding content.
Comment: “Should the equations on page 8 be in the journal equation form and not imbedded with the text?.”
Response: Thanks for your insightful comment. Given that the equations are relatively short, we have not defined them separately as formal mathematical expressions; instead, we have embedded them within the text. If modifications are required in the final version, we will make the necessary changes.
Comment: “varying sig figs in the results, standardize throughout.”
Response: Thanks for your insightful comment. I have normalized the numbers.
Comment: “What is the "so what" of this work? a better way for the computer to recognize a pig in a video? seems like the point that it will help managers id sick pigs and manage without a stronger lead on to how this work will help the industry.”
Response: Thanks for your insightful comment. I'm not sure if I fully understand this comment, but I can give my some thoughts. We focus on real-time instance segmentation. Instance segmentation helps to separate pigs from the complex environment of group raising. This is a basic work for studying individual pigs in group raising pigs. To study the behavior of a certain pig, we need to locate the corresponding pig, whether in an image or in a video, so this work is the basis for further research on other pigs.
Comment: “Why is there a "(a)" after the Figure call out in text?”
Response: Thanks for your insightful comment. I want to differentiate between different images. I can delete it if you feel it's not necessary.
Round 2
Reviewer 1 Report
Comments and Suggestions for Authors
I am glad that my comments were useful and inspired the authors of the article.
Reviewer 3 Report
Comments and Suggestions for Authors
The authors addressed most of my comments.